# Epigenetic and physiological alterations in zebrafish subjected to hypergravity

**Marcela Salazar**[1], **Silvia Joly**[1], **Guillem Anglada-Escudé**[2,3], **Laia Ribas**[1]*

**1** Department of Renewable Marine Resources, Institut de Ciències del Mar—Consejo Superior de Investigaciones Científicas (ICM-CSIC), Barcelona, Spain, **2** Department of Astrophysics, Institut de Ciències de l'Espai—Consejo Superior de Investigaciones Científicas (ICE-CSIC), UAB Campus at Cerdanyola del Vallès, Barcelona, Spain, **3** Institut d'Estudis Espacials de Catalunya–IEEC/CERCA, Gran Capità, 2–4, Edifici Nexus, Despatx 201, Barcelona, Spain

* lribas@icm.csic.es

**Data Availability Statement:** Data of the videos of this manuscript are available in the DigitalCSIC repository: https://doi.org/10.20350/digitalCSIC/16106.

## Abstract

Gravity is one of the most constant environmental factors across Earth's evolution and all organisms are adapted to it. Consequently, spatial exploration has captured the interest in studying the biological changes that physiological alterations are caused by gravity. In the last two decades, epigenetics has explained how environmental cues can alter gene functions in organisms. Although many studies addressed gravity, the underlying biological and molecular mechanisms that occur in altered gravity for those epigenetics-related mechanisms, are mostly inexistent. The present study addressed the effects of hypergravity on development, behavior, gene expression, and most importantly, on the epigenetic changes in a worldwide animal model, the zebrafish (*Danio rerio*). To perform hypergravity experiments, a custom-centrifuge simulating the large diameter centrifuge (100 rpm ~ 3 *g*) was designed and zebrafish embryos were exposed during 5 days post fertilization (dpf). Results showed a significant decrease in survival at 2 dpf but no significance in the hatching rate. Physiological and morphological alterations including fish position, movement frequency, and swimming behavior showed significant changes due to hypergravity. Epigenetic studies showed significant hypermethylation of the genome of the zebrafish larvae subjected to 5 days of hypergravity. Downregulation of the gene expression of three epigenetic-related genes (*dnmt1*, *dnmt3*, and *tet1*), although not significant, was further observed. Taken altogether, gravity alterations affected biological responses including epigenetics in fish, providing a valuable roadmap of the putative hazards of living beyond Earth.

## Introduction

Gravity is a fundamental physical component in the Earth's environment, and it plays a crucial role, in shaping the evolution of all life forms [1]. The comprehension of the effects of this critical environmental factor is required for astronauts and future settlements beyond Earth. Thus, since the beginning of spatial exploration, numerous studies in altered gravity have been performed in both microgravity and hypergravity conditions (reviewed in [2]). While

**Funding:** This study was supported by the Spanish Ministry of Science and Innovation grant 2PID2020-113781RB-I00 "MicroMet" and by the Consejo Superior de Investigaciones Científicas (CSIC) grant 02030E004 "Interomics" to LR. GA-E's work is supported by a Ramón y Cajal 2018 fellowship RYC-2017-22489 from Agencia Estatal de Investigación (AEI). We would like to thanks to JAEICU programme (CSIC) for the grant JAEICU2021-ICE-CSIC to MS. The funders had no role in study design, data collection and analysis, decision to publish, or preparation of the manuscript. In addition, this study was supported by the Spanish government through the 'Severo Ochoa Centre of Excellence accreditation (CEX2019-000928-S), resources from Unidad de Excelencia María de Maeztu CEX2020-001058-M, and the Generalitat de Catalunya/CERCA programme. We also thank the lab technician Gemma Fusté for her essential assistance in fish facilities.

**Competing interests:** The authors have declared that no competing interests exist.

microgravity experiments are complex as they require the presence of samples in space—or in some cases by using microgravity simulators on Earth as clinostats with their technical limitations—[3], hypergravity experiments can be easily simulated on Earth and, consequently, more convenient and considered as ground-based studies.

For many years, hypergravity has been used as a model for understanding the gravity alteration in space. Hypergravity refers to the conditions where the gravity exceeds that on the Earth's surface [4]. These hypergravity studies usually use a centrifuge machine with a large enough radius that shear forces in the sample [5,6]. Hypergravity can induce significant alterations in various physiological systems within the body, and a large literature is found related to physiological alterations. For instance, it has been observed that hypergravity can impact the immune systems of various species throughout evolution, ranging from humans and mammals [7–9], to amphibians [10,11], fish [12], and insects [13,14]. The vestibular system is one of the most studied systems due to its role in maintaining body equilibrium within Earth's gravitational field. This sensory apparatus which coordinates balance and movement is altered under gravitational variations resulting in orientation problems for astronauts [15,16].

Gravity alteration experiments have been conducted using a wide range of animal models, including *Drosophila* [17], *Caenorhabditis elegans* [18], mice and rats [19,20], and fish [21]. Among these, zebrafish (*Danio rerio*), a tropical freshwater fish, belonging to the teleost family, has become an unprecedented tool for research in a variety of fields, such as genetics [22], development [23], toxicology [24], physiology [25], aquaculture [26,27], among others. Its short generation time, the large amount of fertilized eggs, its transparency during early development, the high similarity with human orthologous genes (~70%), and, the availability of genomic resources, make zebrafish an excellent animal research model. Besides the above-mentioned features, zebrafish have an additional advantage due to their physical nature, as it is born and bred in a neutral gravitationally environment, i.e., aquatic, which makes an absence of body weight related to proprio-perceptions, reducing the influence of gravity on supporting tissues and muscle and, higher sensitivity due to relative larger otoliths with differently positioned sacculus-otolith membranes [28]. These characteristics make zebrafish an excellent model for microgravity and hypergravity studies and so a valuable tool in the current space exploration era. In fact, one of the most studied systems in zebrafish is the vestibular system [29–31], where it has been found that microgravity affects otolith development. Further studies showed alterations in genes involved in lens development [32], together with hematopoiesis- and cardiovascular-related genes indicating that short-term hypergravity induced physiological changes in the zebrafish embryos [33].

Epigenetic mechanisms are responsible for the alteration of the final phenotype under environmental pressure, and it is defined as permanent changes in gene expression that occur without modifying the nucleotide sequence of the genome [34,35]. One of the most studied epigenetic mechanisms is DNA methylation which plays a key role in regulating cellular processes in living organisms [36]. This type of modification implies the addition of a methyl group to the 5' position of cytosine (5mC) cytosine-phosphate guanine (CpG) dinucleotides, named as CpG sites [37]. DNA methylation is performed by enzymes known as DNA methyltransferases (dnmts) that catalyze the methylation reactions [38,39]. In mammals, the main dnmts include DNA methylation transferase 1 (dnmt1) responsible for maintaining DNA methylation levels in the cells, and another two, dnmt3a and dnmt3b, for *de novo* DNA methylation [40,41]. To maintain the genomic methylation homeostasis, cells rely on these DNA methyltransferases but also demethylase enzymes. DNA demethylation is a complex process, not fully understood, in which 5mC is converted to 5-hydroxymethylcytosine (5hmC) by the ten-eleven translocation (Tet) family of dioxygenases [42]. Among different tet members, tet1

is the most studied because plays a key role in the demethylation process preventing DNA from methylation maintenance [43].

For the epigenetic mechanism to be stable, three basic components are required: epigenator, epigenetic initiator, and epigenetic maintainer [44,45]. An epigenator is a crucial signal responsible for initiating the intracellular pathway in response to environmental stimuli. It can be any factor or event that triggers the activity of the initiator molecule [46]. Remarkably, any environmental change, regardless of its nature, has the potential to act as an epigenator and produce a signal that persists long enough to exert a significant impact on the epigenetic phenotype [47,48]. Thus, gravity alteration is an epigenator. Although epigenetics has been extensively studied in various research areas over the last two decades, research specifically involving epigenetic modifications in altered gravity conditions remains limited. In *Arabidopsis in vitro* cultures, both microgravity and hypergravity experiments resulted in increased DNA methylation [49]. Transcriptomic studies in rats revealed that genes related to the DNA methylation machinery were altered [50]. Similarly, in cultured human cells, microgravity altered the methylome together with the transcriptome harboring the understanding of molecular events [51]. In 2019, a National Aeronautics and Space Administration (NASA) experiment on twins deciphered multidimensional data after a one-year-long human spaceflight, revealed DNA methylation changes in immune and oxidative stress-related pathways, along with other alterations at various cellular levels (e.g., microbiota, metabolism, transcriptome, or body mass) [52]. Although current data proposed that the adaptation to gravity alterations may proceed through epigenetic changes, more research in this flourishing field needs to be explored before the space environment is a safe place to be. To date, no altered-gravity studies involving zebrafish and epigenetics have been reported. Thus, in order to tackle space-incurred epigenetic disturbances, here examined, for the first time, the epigenetics changes by analyzing the global DNA methylation patterns and the gene expression of three epigenetic-related genes (*dnmt1*, *dnmt3*, and *tet1*). To shed light on the side effects of hypergravity during zebrafish development, fish survival, hatching rate, and physiological traits, were also addressed.

## Materials and methods

### Zebrafish husbandry

Zebrafish (TUE strain) were housed in the animal facilities of the experimental aquariums zone (ZAE) at the Institute of Marine Sciences (ICM-CSIC, Barcelona, Spain). Fish were held in 9 liters (L) tanks on a recirculating system (Aquaneering, San Diego, CA) in a chamber with a photoperiod of 12 h of light and 12 h of darkness, an air temperature of 26 ± 1˚C and a humidity of 60 ± 3%. Physicochemical parameters were monitored daily, staying at appropriate conditions [26,27]: water was maintained at (28 ± 0.2˚C), pH (7.2 ± 0.5), conductivity (750–900 μS) and dissolved oxygen (6.5–7.0 mg/l) with a water pump of 3,000 L/h and a UV light system to eliminate any possible bacteria in the water. Sulfite, sulfate, nitrate, and ammonia quality parameters were checked weekly using commercial kits. Adult fish were fed twice daily, receiving dried food and live *Artemia nauplii* (AF48, INVE Aquaculture, Dendermonde, Belgium).

### Hypergravity device

A custom-made hypergravity centrifuge was performed by using a regular laboratory rotary mixer with a maximum speed of 100 rpm (model: ANR100DE, OVAN laboratory equipment) by adding two perpendicular arms of R = 25 cm ending in two opposite gondolas to support an experimental plate (Fig 1A and 1B). This setup allowed achieving a gravity-like acceleration

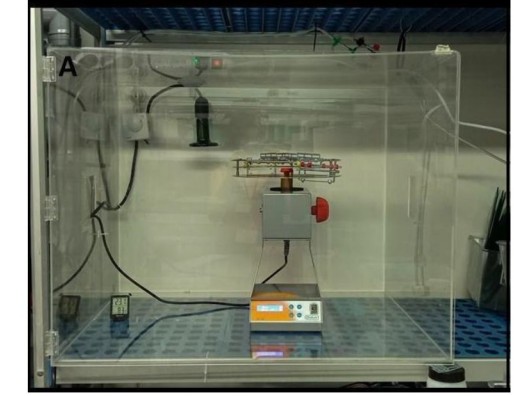

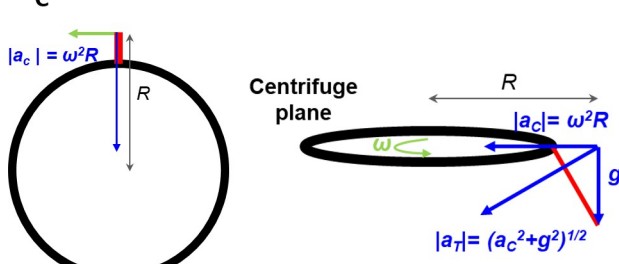

**Fig 1. A)** A custom-made hypergravity centrifuge was used to perform the experiments. **B)** Gondola with the 96 well plate where the larvae were placed during the 5 days of treatment. **C)** Graphical representation of the centrifuge and gondola showing the variables and vectors used to calculate the value of the acceleration (artificial hypergravity) at the center of the gondola. The centrifuge radius, R, is defined as the distance from the center of rotation to the outer edge of the platter. The gondola arm length l was measured from the outer edge of the platter of the centrifuge to the end of the gondola. Both of them (R and l) were measured with a tape measure. The angle, $\theta$, was calculated using taking an image and analyzing it using a program called *Angulus* (DPP v1 2020). r is the gondola radius, v is the tangential velocity due to rotation, $a_c$ is the centripetal acceleration, and $\omega$ is the angular velocity.

of up to ~3 $g$ (where $g$ refers to 9.81 m/s$^2$). To maintain the required environmental conditions for the larvae (27–28˚C and 60–65% humidity) during the hypergravity experiments a methacrylate box was adapted with a heating system using ProClima v8.1.6.1 software (Schneider Electric 2020). These parameters were checked daily. The hypergravity device was set up to 100 rpm (revolutions per minute, or 10.47 rad/s) spin delivering a centripetal acceleration ($a_c$) of 27.4 m/s$^2$ that vectorially added to the existing Earth acceleration ($g$) of 9.81 m/s$^2$, resulting in total acceleration $a_T$ = 29.1 $m/s^2$, which corresponds to 2.96 $g$ (or approximately 3 $g$). The geometry and the justification for these calculations are illustrated in Fig 1C.

## Experimental design

Zebrafish pairs (one female and one male) were bred by transferring them to breeding tanks with a transparent separator overnight. The following day, the separator was removed and fertilized eggs were collected and maintained in Embryo Medium Solution (EMS, Ph = 7,2; NaCl 0.8 g/L, KCl 0.04 g/L, $NaH_2PO_4$ 0.0036 g/L, $H_2KO_4P$ 0.006 g/L, $CaCl_2$ 0.144 g/L, $MgSO_4$ 0.12 g/L, $NaHCO_3$ 0.35 g/L, and 20 μl/L Trypan blue as antifungal). The total number of fertilized eggs was counted to guarantee fertility according to the reference values for this species and fish post-hatch survival agreed with OECD's guidelines for the Fish Sexual Development Test [53].

Fertilized eggs were collected and placed individually into 96-well plates with 250 μl of EMS and were covered with an adhesive sealer to avoid fluid loss. For each biological pair, embryos were distributed in three different plates: the control group was placed inside the methacrylate box, the hypergravity group was placed into the gondolas, and the mock group (MG) was placed in the zebrafish embryonic incubator (28˚C) outside the methacrylate box. The MG was used as an internal control of the standardized embryonic development in this fish species [54].

Two preliminary trials to determine the gravitational force at which we could observe a distinct and measurable impact on zebrafish larvae were conducted by using at speeds of 70 (2 g) and 100 rpm (3 g) by using 22 larvae in both the simulated hypergravity (SHG) and the control groups. During the 70-rpm experiment, no significant responses in the survival or morphology of the zebrafish larvae were observed. However, in the 100-rpm experiment (3 *g*), during the 2- and 3-days post-fertilization (dpf) stages, which coincide with the hatching period, a notable response was observed. Furthermore, the survival rate at 3 *g* decreased by 30% compared to the control, and teratological effects were observed in the larvae.

Upon establishing the gravity level at 3 *g*, the experiment was replicated six times, employing distinct single pairs each time to ensure biological replication. In total, 440 individuals were analyzed, with 220 individuals allocated to the control group and an additional 220 individuals assigned to the hypergravity group. To evaluate the survival and hatching rate, the hypergravity device was stopped for 6–10 minutes every 6 hours during day light (i.e., 10.00 am and 16.00 pm) until 100% of the larvae were hatched (i.e., 3 dpf). After that, the device was stopped every 24 hours. Embryonic and larvae development for each of the six groups three groups was observed by a Leica EZ4 Stereo Microscope (Leica Microsystem Ltd.). After a maximum of 10 minutes, the embryos were placed in their respective experimental conditions until the end of the experiment.

## Ethogram activity

At 5 days of the experiment, the larvae were observed and recorded to assess their ethogram activity. This analysis consisted of identifying three locomotor characteristics: position, movement frequency, and swimming behavior. Table 1 describes each of the observed characteristics for each larva. Support information creating a repository of the videos representing all different locomotor characteristics studied in this manuscript available in the DigitalCSIC repository: https://doi.org/10.20350/digitalCSIC/16106. Larvae were observed and recorded individually for each of the experimental groups with a total of 180 observations (N = 6 biological groups, N = 15 larvae hypergravity, N = 15 larvae control group). Teratologies resulting from the treatment were carefully observed and recorded using the Stereo Microscope (Leica Microsystem Ltd.). To minimize inter-observer error bias, all analyses were performed by the same researcher.

**Table 1. Ethogram description of zebrafish observed in the experiments.**

| Activity | Characteristic | Description |
|---|---|---|
| Position | Horizontal | Normal dorsal position |
| | Horizontal lateral | Lateral up position |
| | Vertical ascendant | The fish's body is in vertical position and its head up |
| | Vertical descendent | The fish's body is in vertical position and its tail up |
| Movement frequency | High | 5 or more movements in 30 seconds |
| | Medium | 2 to 4 movements in 30 seconds |
| | Low | 1 movement in 30 seconds |
| | Static | No movement |
| Swimming behavior | Normal swimming | Normal Swimming |
| | Erratic flotation | Swimming in circles, up and down |
| | Jerky movements | Fast and repetitive movements without displacement |
| | Wrong swimming | Swimming upside down or on their side |

## DNA extraction

A total of 10 larvae for each control and hypergravity group (N = 20) for one biological replicate were digested overnight at 56˚C with a buffer containing 1 μg of proteinase K (Sigma-Aldrich, St. Louis, Missouri) to eliminate proteins. Then, the standard phenol-chloroform-iso-amyl alcohol protocol (PCI 25:24:1) with 0.5 μg ribonuclease A (PureLink RNase A, Life Technologies, Carlsbad, California) was performed to isolate DNA and eliminate RNAs. The quality and quantity of DNA were measured by Qubit (Thermo Fisher Scientific, Waltham, Massachusetts). Isolated DNA samples were stored at −20˚C until further analysis.

## Global DNA methylation analysis

Global DNA methylation was performed in genomic DNA using a 5-mC DNA ELISA kit (Zymo Research, USA) following the manufacturer's protocol and that described in [55]. Briefly, 100 nanograms (ng) of each DNA sample was used for analysis. The standard curve was prepared by mixing negative and positive controls at different proportions. The final methylation concentrations of standards were 0%, 5%, 10% 25%, 50%, 75%, and 100%, respectively. The absorbance was measured at 405 nm using an ELISA plate reader (Infinite® 200 PRO, Tecan™). All samples were analyzed in duplicates. The percentage of 5-mC for unknown DNA samples was calculated using the equation: $\% 5 - mC = e\{(Absorbance - y - intercept)/Slope\}$. Percent 5m-C values were corrected with the zebrafish CpG density according to the manufacturer's instructions. The percentage of CpG was calculated according to the formula presented by Valdivieso et al., 2020 [55], where the latest zebrafish genome from Ensembl was downloaded (www.ensembl.org). Then, the length of the genome (L) = 1,674,207,132 bp was extracted, and the total number of cytosines (C) and the number of CpG dinucleotides (CG) were calculated. Once the total number of C = 306,412,859 and CG = 29,220,867 were obtained from the zebrafish genome, the fold difference of CpG density (total CG genome/L) between the genomes of (*E. coli/D. rerio*) = (0.07472/0.0175) = 4.2811 was calculated. Finally, to obtain the global methylation values, the % 5m-C/CpG density values were multiplied by the value obtained from the total number of C/L zebrafish = 0.1830.

## Gene expression analysis

RNA was individually extracted from 10 larvae in each group (control and hypergravity) for one biological replicate (N = 20 total) with TRIzol (T9424, Sigma-Aldrich, St. Louis, Missouri)

**Table 2. Gene-specific primers used for quantitative PCR.**

| Gene name | Abbreviation | Primer sequence (5'- 3') | Acc. No (GenBank) |
|---|---|---|---|
| Elongation factor 1 alpha | *EFα* | F: CTGGAGGCCAGCTCAAACAT | NM_131263 |
| | | R: ATCAAGAAGAGTAGTACCGCTAGCATTAC | |
| Ribosomal Protein L13a | *RPL13A* | F: TCTGGAGGACTGTAAGAGGTATGC | NM_212784 |
| | | R: AGACGCACAATCTTGAGAGCAG | |
| DNA (cytosine-5-)-methyltransferase 1 | *dnmt1* | F: TCTTCAGCACTACAGTTACCAATCCT | NM_131189 |
| | | R: CGTGCACATTCCCTGACACT | |
| DNA (cytosine-5-)-methyltransferase 3 beta | *dnmt3b* | F: AAGATTTAGGCGTCGGTTTCG | NM_131386 |
| | | R: GTGTCACCCCCTTCAATTAACTG | |
| Ten-eleven-translocation 1 | *tet1* | F: TGACTCACCAGCACTTGAAAAC | KC689999.1 |
| | | R: TTGGTGTCCACATCAGCAGT | |

according to manufactured procedures. RNA pellets were suspended in 25 μl DEPC–water and kept at -80˚C. RNA concentration was determined by ND-1000 spectrophotometer (NanoDrop Technologies) and RNA quality was checked on a 1% agarose/formaldehyde gel. Following supplier protocols, 100 ng of total RNA for each sample was treated with DNAse I, Amplification Grade (Thermo Fisher Scientific Inc., Wilmington, DE, USA) and retrotranscribed to cDNA with Transcription First Strand cDNA Synthesis kit (Roche, Germany) with Random hexamer (Invitrogen, Spain).

Quantitative Polymerase Chain Reaction (qPCR) was performed using synthesized cDNA, previously diluted 1:10 with DNase-free water, 5 μl of 2X qPCRBIO SYBR Green Mix Lo-ROX (PCR Biosystems), 0.5 μL of each forward and reverse primers, and 2 μL of DNase free water. qPCR was carried out in technical triplicates for each sample. The conditions in the thermocycler were as follows: initial denaturation for 3 min at 95˚C, 39 cycles of 10 s at 95˚C, 30 s at annealing temperature; followed by melt curve analysis (65˚C–95˚C at 0.5˚C/5 s) to verify amplification of a single product. The dissociation step, primers efficiency curves, and PCR product sequencing confirmed the specificity of each primer pair. All primers efficiencies ranged between (95–104%). Primer sequences were designed using Primer3web v4.1.0 [56] and further information is found in Table 2.

## Statistical analysis

All statistical analyses were conducted using R (v. 1.1.456). The *homogeneity of variances* was assessed using *Levene's* test, followed by a *Shapiro-Wilk* test to check for the normality of the data. For normally distributed data, we performed an *ANOVA*, and for non-normal data, we utilized non-parametric tests such as *Kruskal-Wallis*. Tukey's test was used to perform *post hoc* multiple comparisons.

To assess differences between groups and categories in the ethology, we employed the Chi-square test. Furthermore, a t-test for methylation data was performed to determine if there is a significant difference in methylation levels between hypergravity and control groups, the data is presented as percentage of CpG methylation in a lollipop plot.

Data obtained from qPCR were collected by SDS 2.3 and RQ Manager 1.2 software. For each sample, the relative quantity (RQ) values of *dnmt1*, *dnmt3*, and *tet1* gene marks were used to normalize against the geometric mean value of two internal control genes: *rpl13a* (ribosomal protein L13A) and *efα* (elongation factor α) [57], and the fold change was calculated using the $2^{\Delta\Delta Ct}$ method [58]. Ten samples were used for each condition (control and hypergravity). Data are shown as mean ± SEM of fold change using control values set at 1.

Relative Expression = 2^(-ΔCt). Statistical significance for all evaluated parameters was considered at *P<0.05*. Graphs were created using the ggplot2 package (v.3.1.0) by [59].

## Ethics statement

The procedure for using zebrafish in this study was conducted following approved guidelines by the Bioethical Committee of the Generalitat de Catalunya (reference code 9977) and the Spanish National Research Council (CSIC) Ethics Committee (reference code 1166/2021). In the present study, the fish rearing and maintenance were following the European regulations of animal welfare (ETS N8 123, 01/01/91 and 2010/63/EU). Fish facilities in the ICM were validated for animal experimentation by the Ministry of Agriculture and Fisheries (REGA number ES080190036532).

## Results

### Survival and hatching

The survival of zebrafish embryos and larvae exposed to hypergravity was determined every dpf during the experiment. As shown in Fig 2, the survival was significantly affected at 2 dpf decreasing embryonic survival. In contrast, no differences were found during larva development until the end of the experiment between treatment groups. The hatching rate was not significantly affected by the hypergravity condition, occurring between the second and third day both in control and in hypergravity (Fig 3), as it normally occurs in zebrafish. Most of the hatching occurred at 58 hours post fertilization (hpf) in both groups and at 72 hpf the hatching was completed for all the experimental individuals in both groups. However, at 52 and 64 hpf, the hatching rate was decreased in hypergravity when compared to control, from 15 to 10% and 90 to 80% in control and hypergravity, respectively.

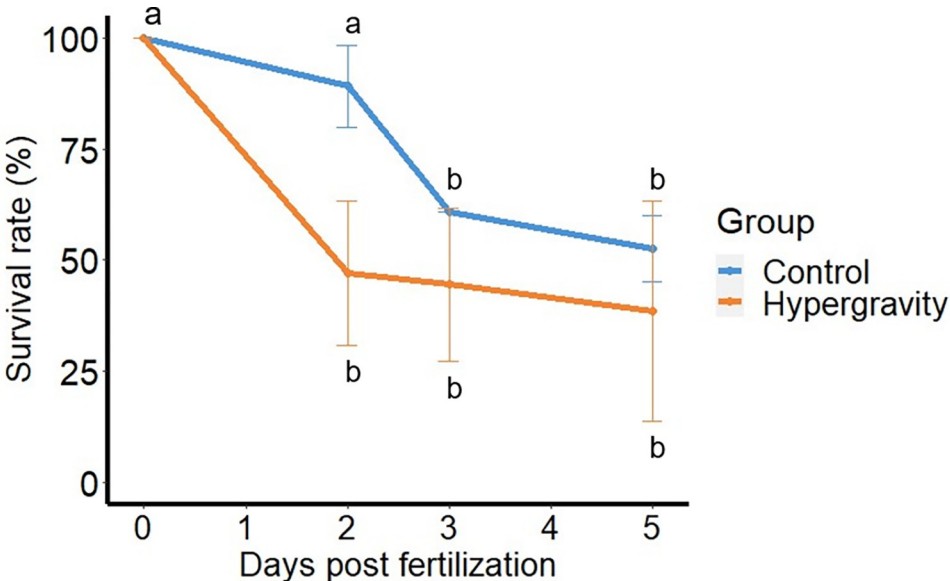

**Fig 2. Survival rates of control and hypergravity zebrafish larvae during 5 days of treatment.** Each data point shows the mean ± SE of six independent groups with a total number of 220 individuals per condition (control and hypergravity). The cumulative survival with different letters indicates a significant difference (P < 0.005) according to the Least significant difference (LSD) test.

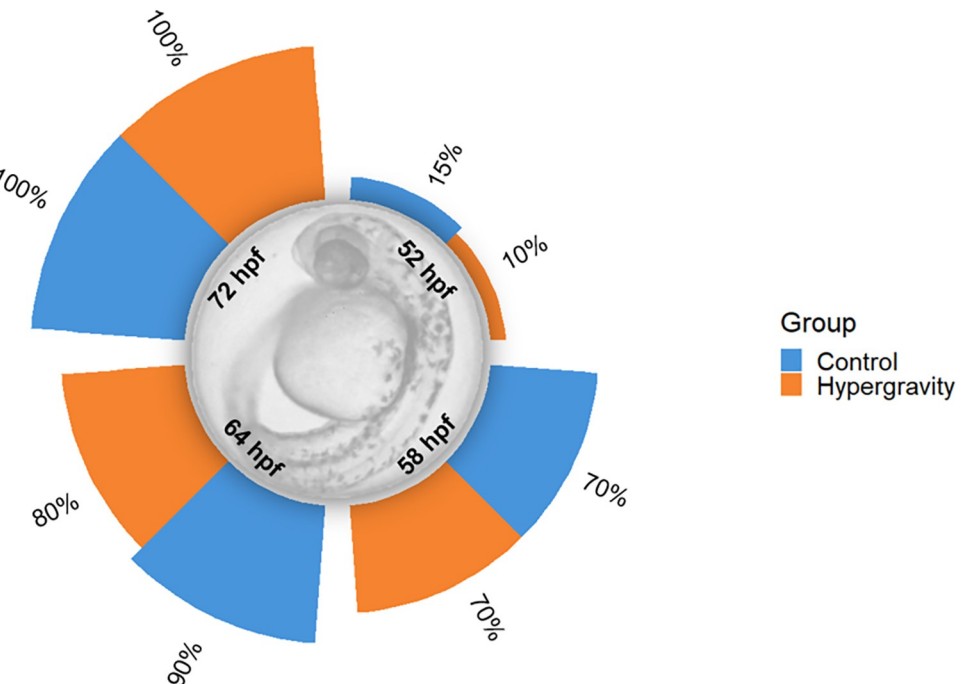

**Fig 3. Hatching rate of larvae at 2- and 3-days post fertilization treated with hypergravity compared with the control group.** Six biological replicates were made, with a total number of larvae of 220 and 220 in control and hypergravity, respectively. Data are presented as percentage ± standard error of the mean (SEM). Normality was evaluated with a Kolmogorov–Smirnov test, and Levene's test was used to assess homoscedasticity of variances. No differences between groups were found according to Least significant difference (LSD) test.

## Ethogram analysis

The locomotor activities of exposed larvae were evaluated after 5 days of hypergravity. The observation of the fish's position, movement frequency, and swimming behavior showed high significance in almost all the studied characteristics (Figs 4 and 5). For control, 75.40% of the larvae were found in horizontal (normal position) whereas only 18.26% of the larvae in hypergravity showed this same position (Fig 4A). The most observed position for hypergravity-exposed larvae was the vertical ascendant position with 47.10% while in control only 11.50% of the individuals presented this position. Vertical descendent position was present in 13.50% of larvae subjected to hypergravity while none of the control was observed.

More than 90% of the larvae in the hypergravity had no movement frequency (59.6%) or low movement frequency (30.8%) and only 3.8% and 5.8% had high or medium movement frequency. In contrast, ~51% of the larvae in control showed high or medium movement frequency and the other half presented low (16.4%) or static (32.80%) movement frequency (Fig 4B).

Hypergravity was able to altered significantly the swimming behavior by increasing jerky movements (32.70%) and wrong swimming (7.70%) and, decreasing larvae with normal swimming when compared to control conditions (Fig 4C). No significance was found for Erratic swimming individuals between groups.

Some zebrafish larvae showed teratologies after the hypergravity treatment. These teratologies consisted of four major types based on their position: body curvature, tail curvature, abnormal eye size, and overall body deformation (Fig 5).

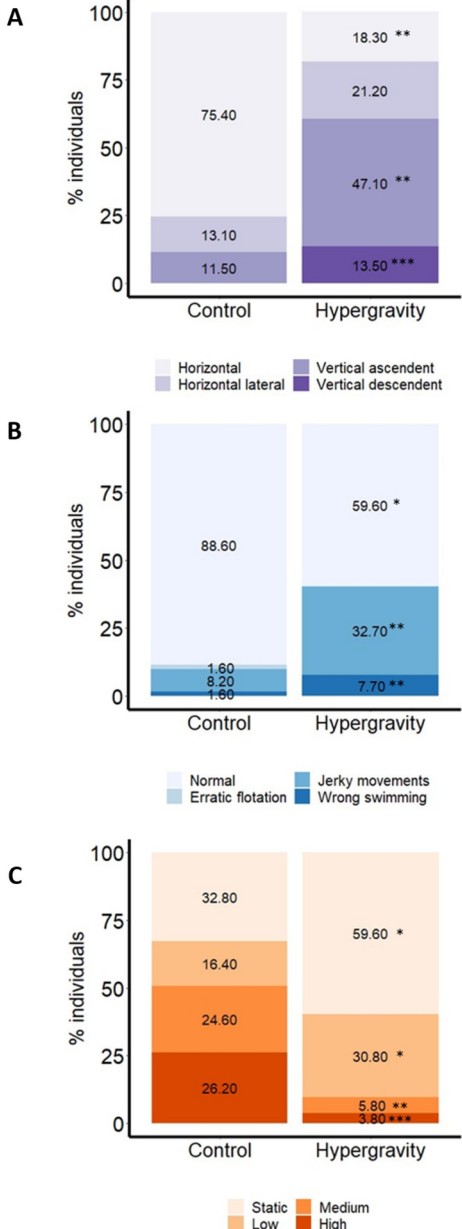

**Fig 4.** Ethogram analysis consisted on identifying three locomotor characteristics: Position (**A**), movement frequency (**B**) and swimming behavior (**C**) of larvae of zebrafish at 5 days post fertilization (dpf). Bar graphs representing the percentage of individuals in control and hypergravity conditions. Six biological replicates were observed with a total number of 90 individuals in the hypergravity group and 90 in the control. The different colors represent the assessed features in each parameter. The data are presented as percentages of individuals. To evaluate significant differences, we performed a Chi-square test. * = $P < 0.05$; ** = $P < 0.01$; *** = $P < 0.001$.

## Methylation alterations

Hypergravity exposure was able to alter significantly the global DNA methylation levels between the control and the hypergravity group (*t*-test, *P < 0.007*, Fig 6). Larvae in hypergravity presented hypermethylation levels of the global DNA when compared with the control group (percent CpG mean was 21.04% and 26.53%, in control and hypergravity groups, respectively).

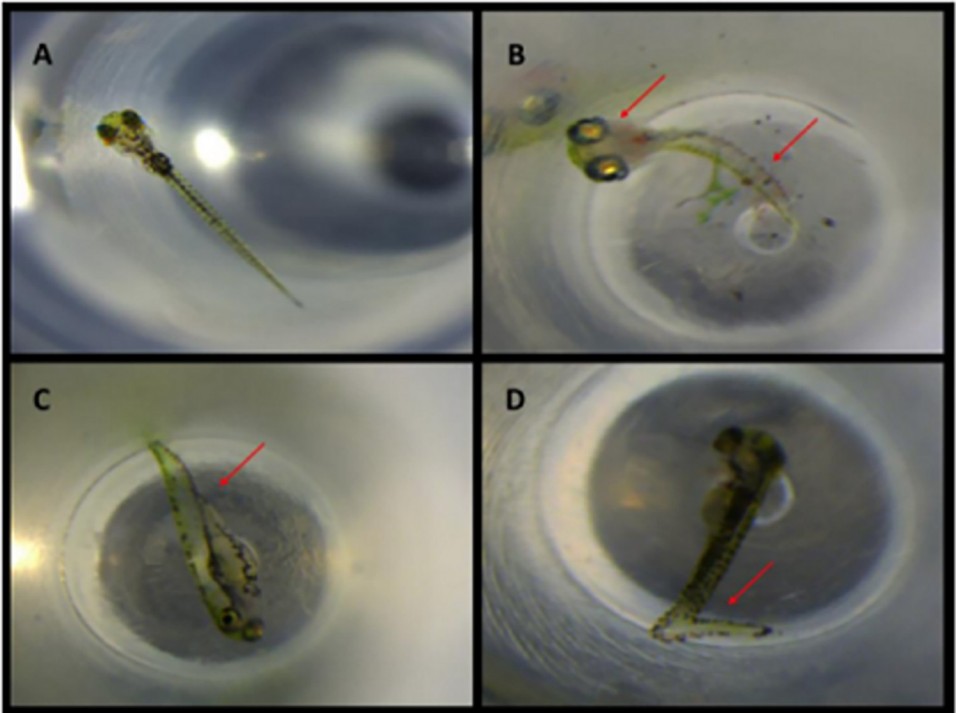

**Fig 5. Teratology was observed at 5 days post fertilization (dpf) in zebrafish larvae exposed to ~3 *g* hypergravity from 0 to 5 dpf.** Positions of the larvae are shown in the figure: horizontal (A), vertical ascendant (B); horizontal lateral (C); and vertical descendent (D). Teratologies included four major types: body curvature (B), abnormal eye size (B), overall body deformation (C) and tail curvature (D).

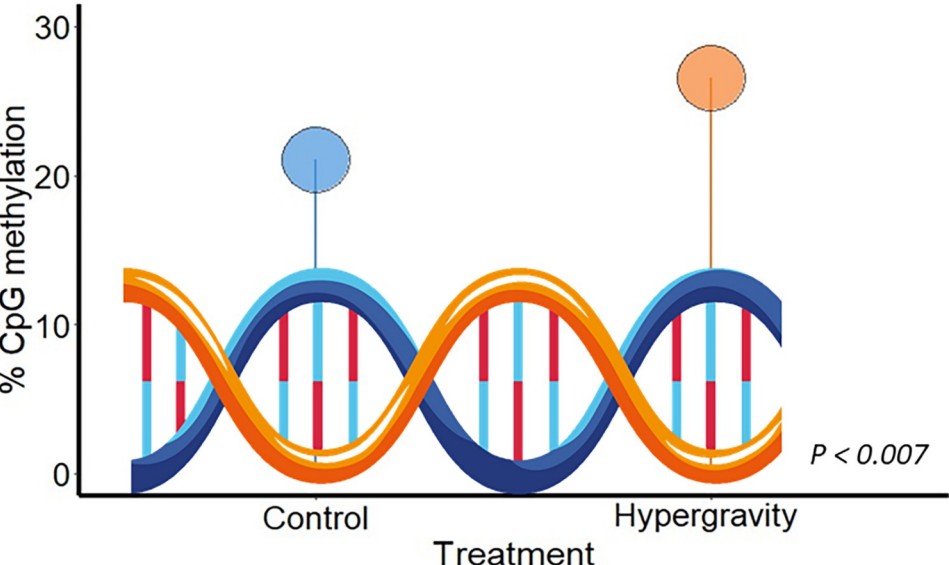

**Fig 6. Global DNA methylation in zebrafish larvae after 5 days of ~3 *g* hypergravity.** N = 10 larvae per group (control and hypergravity). The statistical analysis was conducted using a two-tailed t-test (p < 0.05), demonstrating a significant difference in DNA methylation levels between the control and hypergravity conditions. Each lollipop represents the percentage of methylation in control and hypergravity conditions.

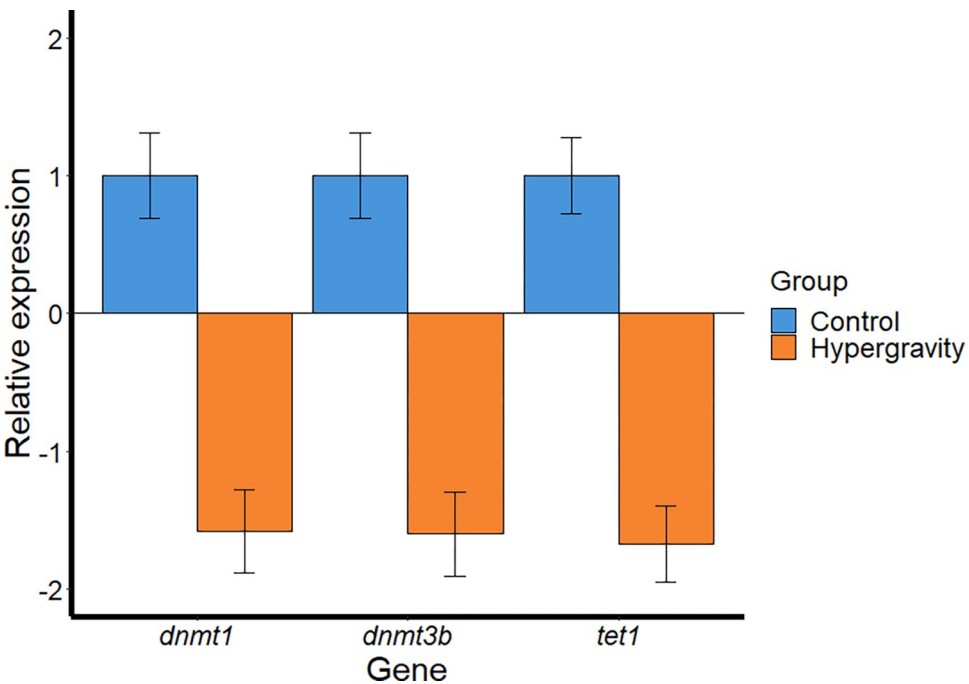

**Fig 7. Expression of three epigenetic-related genes in zebrafish larvae after 5 days of ~3 *g* hypergravity.** Data are shown as mean ± SEM of fold change Relative Expression = 2^(-ΔCt) using control values set at 1. N = 10 larvae per group (control and hypergravity). No significant differences were found.

## Gene expression response

Hypergravity exposure of the zebrafish larvae caused downregulation of the three studied epigenetic markers. The fold change was -1.6, -1.5, and -1.6 for *dnmt1*, *dnmt3b*, and *tet1*, respectively, although without significance (Fig 7).

## Discussion

### Hypergravity influenced larvae survival but did not alter the hatching rate

Very limited information exists on the impact of altered gravity on fish survival and hatching success. In the present study, hypergravity was able to decrease embryonic survival at 2 dpf just before larvae started to hatch. In contrast, the hatching rate was not significantly affected by hypergravity, although it slightly decreased the hatching rate at 52 and 68 hpf. Almost 40 years ago, the first fish on the space base station Skylab 3, mud minnow (*Fundulus heteroclitus*) eggs were able to hatch successfully with a rate of 96% indicating that gravity alteration did not affect the hatching rate [60,61]. Gravity alteration did not affect medaka (*Oryzias latipes*) as embryos hatched after being fertilized in space but also those space-fertilized eggs after being sent back to Earth after 3 days of landing [62–64]. Similarly, analogous space-related experiments in European sea bass (*Dicentrarchus labrax*) and meager (*Argyrosomus regius*) showed no significant difference after exposing eggs to a simulated spacecraft launcher vibration [65]. Thus, overall current available data indicate that hatching might be not sensible to the environmental stress caused by gravity alterations, probably due to the natural protection of the chorion to external stimulus.

## Hypergravity affected the morphology and behavior

In the zebrafish model, larvae exhibit mature swimming between 4–5 dpf and respond to visual and stress stimuli increasing the movement [66]. Zebrafish are sensitive to a range of external stimuli such as olfactive, sensitive, vestibular inputs, heat, and vision and many studies in zebrafish showed that swimming is altered under gravity modification [15,67,68]. In the present study, after hypergravity exposure, fish position, movement frequency, and swimming behavior showed drastic significant changes. Swimming behavior was affected by the increase in gravity and larvae showed atypical displacements such as jerky movements and wrong swimming. Most of the tested individuals subjected to hypergravity-treated were positioned in vertical ascendants and a static or with low movement, indicating the environmental stress impact by the hypergravity environment. In space, mummichog fish swam in elongated loops, called a looping response, during the first three days, as they were disoriented because the vestibular system was affected [69]. Looping appeared to be the fish's equivalent of space sickness, although it gradually disappeared as the fish learned how to orient themselves [70]. Thereafter, fish living in space swam in regular patterns, oriented by artificial light, and even were able to mate naturally [71].

Thigmotaxis is a valid index of anxiety in which an animal prefers to be closed to the walls in a vertical position and it is evolutionarily conserved across different species, such as fish, rodents, and humans [66,72,73]. In fact, the vestibular system is responsible for the fish orientation and it is directly affected by gravitational changes [28,74]. A complete map of the gravity-sensing system straddling from the inner ear to the brainstem was described [75]. In fish, otoliths, the inner ear heavy stones, are responsible for fish orientation, and their growth was targeted in space-related research [30,76]. Hypergravity experiments in zebrafish and Cichlid fish embryos (*Oreochromis mossambicus*) showed an increase in the otolith's growth [77–80] while microgravity on space flight or by clinostats yielded an opposite growing effect in swordtail (*Xiphophorus helleri*), in cichlid fish larvae [81,82] and in zebrafish embryos [30]. Generation of a mutant medaka strain ha (genotype ha/ha) with an absence of otoliths, decreased the sensitivity to gravity after microgravity and parabolic flights [83].

After three weeks of hypergravity experiment with cichlid fish (*Oreochromis mossambicus*) larvae, the morphogenetic development and the swimming were not affected. However, when the centrifuge treatment stopped, looping and spinning movements appeared [84]. However, within hours, this kinetic behavior disappeared. Similar results were observed in the present study because after some days of hypergravity treatment the abnormal movement disappeared and fish developed normally (Salazar-Moscoso, personal comments). Similarly, medaka fish had a looping response during three days after landing on Earth, after that, they swam properly [71]. Overall results may indicate that zebrafish are able to adapt to the novel environments thanks to vestibular neuronal system plasticity [85,86].

Some teratologies were observed due to hypergravity, such as body curvature or body deformation. The data presented here aligns with previous findings in zebrafish subjected to similar hypergravity conditions (3 g, from 0 to 5 dpf). These zebrafish exhibited morphometric changes, including an enlarged head and increased cranial bone formation. These findings supported that hypergravity can induce consistent effects on the developmental processes in zebrafish [87].

## Genome-wide DNA methylation levels were altered

One of the most important findings of this study was that hypergravity exposure of zebrafish during 5 days, regulated the epigenetic events by hypermethylation genome-wide DNA levels. Few studies addressed the epigenetic changes in the genome due to gravity alterations, in

particular to hypergravity. To date, and based on our knowledge, this is the first time reported in fish.

DNA methylation levels were altered in response to long-term isolation experiments, for example in mice in the SpaceX-4 mission increasing the total genome methylation [88]. DNA methylation dynamics were analyzed in long-term isolation of simulated space travel in the blood of the crew of the Mars-500 mission, identifying six significant epigenomic patterns at post-isolation recovery [89]. The functions of these DNA methylation patterns were mostly related to the immune-system and tumors but also epigenetic genes related to glucose and mood-state disturbance were observed [90]. As expected, by analyzing epimarkers for aging, the simulated space travel was associated with significant decreases in epigenetic aging meaning that the stress produced by the travel decreased the biological age of the crew [91]. In mice, epigenetic memory was analyzed by the retinal profiles of animals exposed to microgravity and irradiation four months after the exposition [92]. Their results revealed a crosstalk between the epigenome and the transcriptome as 23 potential biomarkers genes related to retinal function and inflammatory response showed significant changes [92]. In this sense, cultured human lymphoblastoid cells revealed that microgravity induced a ~60% hypomethylation and ~92% hyperhydroxy methylated regions of the methylome together with 370 transcripts associated with crucial biological processes [51].

In fish, it is known that the environment-epigenetic interactions are responsible for determining the phenotype in those animals growing in artificial and cultured environments [93,94]. In the last decade, many studies on fish recalled the importance of epigenetics, both during early development and adulthood in cultured conditions. For example, high temperature—the most studied environmental factor in fish—, suffered during larvae stages in the hatcheries, was responsible for masculinizing populations through epigenetic events [95,96]. These epigenetic events that occurred by high temperatures were inherited by the following generation (F1) [55] evidencing the importance of epigenetics, not only for the current generation but for the future. Another environmental factor, the density—the number of fishes in a tank—, caused masculinization in zebrafish [97] revealing some epigenetic alterations in the fish gonads [98]. Additionally, environmental-epigenetics interactions after immune stimulation during sex differentiation in zebrafish showed an alteration of the methylation of some immune-related genes [99,100]. The present data indicated that the epigenetic machinery was reprogrammed by gravity alterations, and although much data needs to be performed, particularly whether zebrafish would be used as a space-related animal model, the underlying epigenetic alterations need to be considered to fully comprehend the gravity-epigenome interactions. In this sense, higher genome coverage techniques, such as whole-genome bisulfite sequencing (WGBS) or a single-nucleotide resolution Nanopore sequencer, would bring better comprehension of the epigenetic dynamics under gravity alterations.

## Expressions of dnmt1, dnmt3 and tet1 downregulated without significance

Epigenetics are dynamic throughout the life of an organism and are implicated in many developmental processes, cell differentiation, genomic imprinting, and modulating gene expression [101,102]. Epigenetic mechanisms are responsible for permanent heritable alterations in cellular gene expression. To gain insight into the molecular events underlying hypergravity's effects, we performed an analysis of three genes related to epigenetic regulation. After 5 days of treatment, there was an inhibition of *dnmt1*, *dnmt3*, and *tet1* expression although not significant. The inverse correlation between the hypermethylation of the DNA levels found in the same experiment with the lower expression of these epigenetic-related genes, was in accordance with the classical dogma; higher DNA methylation is associated with the inhibition of the gene

transcription machinery [103,104]. Nevertheless, it is currently accepted that DNA methylation are dynamic and complex process in which many genomic elements contribute to transcriptional regulation: exons [105], gene body [106], introns [107], and, post transcription modifications [108].

Quite extensive bibliography refers to gene regulation in space exploration-related studies. In a recent review, almost 200 articles were included and the alteration of genes in many organisms and biological systems were described [109]. Thus, reflecting the importance of the knowledge of the consequence of gene expression to tackle the risk to health for the astronauts and alive organisms. In zebrafish larvae, the gene expression in hypergravity (24 h, 3g) revealed differential expression of genes involved in the development and function of the skeletal, muscular, nervous, endocrine, and cardiovascular systems [87]. Simulated microgravity affected the expression of some genes related to fish musculoskeletal, cardiovascular, and nuclear receptor systems [21] and the immune system in response to a viral response [110]. In contrast, the number of articles addressing the expression of epigenetic-related genes is much less extent, not only in fish, but also in other animals and humans, and sometimes contradictive. Likewise to the data obtained in this study, did not change the expression of TET1 and TET3 in cardiac and lung of mice 37 days onboard of the American International Space Station (ISS) of SpaceX-4 mission [88]. DNMT1, DNMT3a, and DNMT3b decreased at 7 days in human T-lymphocytes microgravity exposed [111] and a transcriptomic study of pregnant rats subjected to spaceflight showed that DNMT1 was downregulated while DNMT3a was upregulated [50]. In addition, mutations in the blood samples of space shuttle astronauts were identified in the DNMT3A and TP53 genes [112]. Overall data indicate that epigenetic-related genes are altered due to gravitational changes, but more research needs to be performed to fully understand the crosstalk between the epigenome and the transcriptome.

## Conclusions

This study presents the first evidence of epigenetic impacts on the DNA methylation levels in zebrafish subjected to hypergravity during early development. Although not statistically significant, there was a noticeable downregulation tendency observed in three epigenetic-related genes' expression. Furthermore, the survival rate decreased two days after the treatment, while the hatching rate remained unaffected by hypergravity. In contrast, physiological traits (position, movement frequency, and swimming behavior) of the larvae were drastically affected, accompanied by the observation of some teratologies.

The presented data and experiments explore the new domain of how altered gravity impacts development in living models by, for the first time, looking into epigenetic effects in fish. Future experiments in space shall shed some light on whether the development of adult and fertile animals (and eventually humans) could develop in space or other planetary bodies.

## Acknowledgments

We would like to thanks to JAEICU programme (CSIC) for the grant JAEICU2021-ICE-CSIC to MS.and the Generalitat de Catalunya/CERCA programme. We also thank the lab technician Gemma Fusté for her essential assistance in fish facilities. In addition, we would like to thank

## Author Contributions

**Conceptualization:** Guillem Anglada-Escudé, Laia Ribas.

**Data curation:** Marcela Salazar.

**Formal analysis:** Marcela Salazar.

**Funding acquisition:** Guillem Anglada-Escudé, Laia Ribas.

**Investigation:** Marcela Salazar, Guillem Anglada-Escudé, Laia Ribas.

**Methodology:** Marcela Salazar, Silvia Joly, Laia Ribas.

**Project administration:** Laia Ribas.

**Resources:** Laia Ribas.

**Supervision:** Laia Ribas.

**Validation:** Silvia Joly.

**Visualization:** Laia Ribas.

**Writing – original draft:** Marcela Salazar, Laia Ribas.

**Writing – review & editing:** Guillem Anglada-Escudé, Laia Ribas.

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
