## [Decision Letter · Decision Letter 0]

21 Nov 2023

PONE-D-23-33854Epigenetic and physiological alterations in zebrafish subjected to hypergravityPLOS ONE

Dear Dr. Ribas,

Thank you for submitting your manuscript to PLOS ONE. After careful consideration, we feel that it has merit but does not fully meet PLOS ONE’s publication criteria as it currently stands. Therefore, we invite you to submit a revised version of the manuscript that addresses the points raised during the review process.

The reviewers raised several concerns in this paper, which should be addressed in the revised version.

We look forward to receiving your revised manuscript.

Kind regards,

Hiroshi Kaji

Academic Editor

PLOS ONE

Journal Requirements:

"This study was supported by the Spanish Ministry of Science and Innovation grant 2PID2020-113781RB-I00 “MicroMet” and by the Consejo Superior de Investigaciones Científicas (CSIC) grant 02030E004 “Interomics” to LR. GA-E's work is supported by a Ramón y Cajal 2018 fellowship RYC-2017-22489 from Agencia Estatal de Investigación and JAEICU2021-ICE-CSIC to MS."

"This study was supported by the Spanish Ministry of Science and Innovation grant 2PID2020-113781RB-I00 “MicroMet” and by the Consejo Superior de Investigaciones Científicas (CSIC) grant 02030E004 “Interomics” to LR. GA-E's work is supported by a Ramón y Cajal 2018 fellowship RYC-2017-22489 from Agencia Estatal de Investigación (AEI), resources from Unidad de Excelencia María de Maeztu CEX2020-001058-M and the Generalitat de Catalunya/CERCA programme. In addition, this study was supported by the Spanish government through the ‘Severo Ochoa Centre of Excellence accreditation (CEX2019-000928-S) and JAEICU2021-ICE-CSIC to MS. We thank the lab technician Gemma Fusté for her essential assistance in fish facilities."

"This study was supported by the Spanish Ministry of Science and Innovation grant 2PID2020-113781RB-I00 “MicroMet” and by the Consejo Superior de Investigaciones Científicas (CSIC) grant 02030E004 “Interomics” to LR. GA-E's work is supported by a Ramón y Cajal 2018 fellowship RYC-2017-22489 from Agencia Estatal de Investigación and JAEICU2021-ICE-CSIC to MS."

Reviewers' comments:

Reviewer's Responses to Questions

**Comments to the Author**

1. Is the manuscript technically sound, and do the data support the conclusions?

Reviewer #1: Partly

Reviewer #2: Partly

2. Has the statistical analysis been performed appropriately and rigorously? 

Reviewer #1: No

Reviewer #2: No

3. Have the authors made all data underlying the findings in their manuscript fully available?

Reviewer #1: Yes

Reviewer #2: No

4. Is the manuscript presented in an intelligible fashion and written in standard English?

Reviewer #1: Yes

Reviewer #2: No

5. Review Comments to the Author

Reviewer #1: This manuscript addresses the effects of hypergravity on developmental, behavioral, gene expression, and epigenetic changes in zebrafish. The authors showed significant hypermethylation of the genome and downregulation (not significant) of gene expression levels of three genes under hypergravity for 5 days. This study is important for understanding the effects of hypergravity in DNA methylation level, but needs to be modified before publication in Plos One.

1. Figure 4 and Table 1 show the identification of three locomotor aspects. It could be more easily understood if video data or other means were used to demonstrate these typical behaviors.

2. Figure 5 shows various phenotype under hypergravity. Each is assumed to have different conditions, such as tissue inflammation and damage. Did authors perform the histological analysis? And how did the authors evaluate gene expression and genomic methylation in zebrafish exhibiting these various phenotypes?

3. The authors attempt to show in only one figure (Figure 6) the overmethylation of the zebrafish genome under 5 days of hypergravity. The results of the comprehensive analysis should be evaluated from various angles.

4. Figure 7 describes that there was a decrease in the expression of three genes, but it was not significant. Were significant decreases in gene expression also seen in other genes?

5. Are there any specific sites of DNA methylation under hypergravity condition?

6. Is "Hipergravity" misspelled in figure 6?

Reviewer #2: In this study, Salazar et al. studied the effects of hypergravity on development, behavior, gene expression, and epigenetic alterations in zebrafish. The authors revealed that hypergravity decreased survival, but not hatching rate at 2 days post fertilization. Hypergravity induced physiological and morphological changes including posture, movement frequency, and swimming behavior. Moreover, hypergravity increased DNA methylation and tended to decrease epigenetic-related genes, such as dnmt1, dnmt3, and tet1, without statistical significance in the zebrafish larvae. However, there are several issues with the manuscript. The details are attached below.

Major points

1. Please provide a rationale for used 3 g hypergravity condition in the present study.

2. The detailed methods for analysis of qPCR should be added in Methods section. Did the authors calculated data using ΔΔCt methods? Which endogenous control was used?

3. Although Table 2 showed primer sequence for ef1α and rpl13a, the authors did not show those quantitative data.

4. Fig. 6: According to figure legend, data were expressed as boxplot with thick line indicating median. However, figure 6 were not expressed boxplot. Please correct figure 6 and its figure legend.

5. Statistical analysis should be correct. Although the authors revealed that Shapiro-Wilk test was used to check for normality in Methods section, normality was evaluated with a Kolmogorov-Smirnov test in Fig. 3. The authors used student t-test for assessment of differences in Fig. 6, but there is no description regarding t-test in Methods section. It is not clear which statistical was used in Fig. 7.

6. PLOS authors have the option to publish the peer review history of their article (what does this mean?). If published, this will include your full peer review and any attached files.

Reviewer #1: **Yes: **Masahiro Chatani

Reviewer #2: No

---

## [Author Response · Author response to Decision Letter 0]

14 Feb 2024

Reviewer #1: This manuscript addresses the effects of hypergravity on developmental, behavioral, gene expression, and epigenetic changes in zebrafish. The authors showed significant hypermethylation of the genome and downregulation (not significant) of gene expression levels of three genes under hypergravity for 5 days. This study is important for understanding the effects of hypergravity in DNA methylation level, but needs to be modified before publication in Plos One.

1. Figure 4 and Table 1 show the identification of three locomotor aspects. It could be more easily understood if video data or other means were used to demonstrate these typical behaviors.

Thanks, for the comment. We have added new support information creating Dataset 1 (https://drive.google.com/drive/folders/1PNgmMw1a-Nh0kWXqsyCIH3FOIeCAuOJe?hl=es) which are the videos exhibiting every morphological characteristic observed in both the control and hypergravity groups of zebrafish larvae.

2. Figure 5 shows various phenotypes under hypergravity. Each is assumed to have different conditions, such as tissue inflammation and damage. Did the authors perform the histological analysis? And how did the authors evaluate gene expression and genomic methylation in zebrafish exhibiting these various phenotypes?

Thanks for pointing this out. Unfortunately, histological analysis was not performed because we used the whole larvae for the DNA and RNA extractions, and therefore no more tissues were available. 

Gene expression and methylation analysis were randomly conducted within the groups; 15 samples were ethologically evaluated from each experiment, and 10 of these were randomly selected for the aforementioned studies. We decided not to select any specific phenotype as this could bias the result. However, after evaluating our raw data from the experiment, we have created the following table with the selected phenotypes of those larvae used for molecular analysis.

Teratologies Control Hypergravity

Normal body shape 12 0

Overall body deformation 1 6

Tail anomalies 1 0

Body curvature 1 9

3. The authors attempt to show in only one figure (Figure 6) the over methylation of the zebrafish genome under 5 days of hypergravity. The results of the comprehensive analysis should be evaluated from various angles. 

The reviewer is right with the suggestion. In the current form of the study, we aimed to explore, for the first time, the epigenetic changes that hypergravity could trigger in zebrafish larvae. As the findings are promising, future studies in our lab will be addressed towards a better comprehension of the epigenome modifications from various angles. We are currently conducting RNA-seq from a similar experiment hoping that they will untangle molecular events by gene expression modifications. 

4. Figure 7 describes that there was a decrease in the expression of three genes, but it was not significant. Were significant decreases in gene expression also seen in other genes?

We did not evaluate other genes because we wanted to assess these important genes related to methylation. Based on our experience in the lab, we used them as epimarkers for gene expression studies related to methylation alterations (Ribas et al 2017 https://doi.org/10.1186/s13072-017-0168-7; Valdivieso et al 2023 DOI: 10.3390/ijms242116002). However, as previously mentioned, we are working in the lab on RNA-sequencing technologies and we hope we will be able to identify differential gene expression. 

5. Are there any specific sites of DNA methylation under hypergravity conditions?

For the current study on Global DNA methylation (Zymo Research), the methodology utilized unfortunately lacks the capability to discern specific methylation sites. Despite this limitation, we are actively engaged in the experimentation and evaluation of an alternative technique for our upcoming study. In particular, we are exploring the application of the MinION, a cutting-edge technology that holds promise for providing insights into the methylation patterns. This innovative approach is expected to enhance our ability to pinpoint and analyze specific methylation sites with greater precision, thereby contributing to a more comprehensive understanding of the underlying molecular mechanisms.

6. Is "Hipergravity" misspelled in figure 6?

Thanks, the figure has been amended

Reviewer #2: In this study, Salazar et al. studied the effects of hypergravity on development, behavior, gene expression, and epigenetic alterations in zebrafish. The authors revealed that hypergravity decreased survival, but not hatching rate at 2 days post fertilization. Hypergravity induced physiological and morphological changes including posture, movement frequency, and swimming behavior. Moreover, hypergravity increased DNA methylation and tended to decrease epigenetic-related genes, such as dnmt1, dnmt3, and tet1, without statistical significance in the zebrafish larvae. However, there are several issues with the manuscript. The details are attached below.

Major points

1. Please provide a rationale for used 3 g hypergravity condition in the present study.

Thanks for your comments. We performed two pilot studies before selection 3g hypergravity as the experimental conditions for our experiments. The explanation concerning the selection of 3g have been incorporated into the manuscript.

2. The detailed methods for analysis of qPCR should be added in Methods section. Did the authors calculated data using ΔΔCt methods? Which endogenous control was used?

Yes, we used the ΔΔCt method with two endogenous genes, RPL13A (Ribosomal Protein L13A) and EFα (Elongation Factor α). We have already amended the text accordingly.

3. Although Table 2 showed primer sequence for ef1α and rpl13a, the authors did not show those quantitative data.

These genes were the endogenous controls that were used to normalize the target genes to evaluate the expression making a qualitative study. This method allows to study of gene expression based on qualitative data all the values normalized to their endogenous genes to later compare them with their corresponding controls. Thus, these analyses are not quantitative, in compared to using a specific probe for each gene by qPCR. We hope that adding information to the text as mentioned in the previous comment makes the analysis clearer. 

4. Fig. 6: According to figure legend, data were expressed as boxplot with thick line indicating median. However, figure 6 were not expressed boxplot. Please correct figure 6 and its figure legend.

Thanks, the text in the legend has been amended accordingly. 

5. Statistical analysis should be correct. Although the authors revealed that the Shapiro-Wilk test was used to check for normality in the Methods section, normality was evaluated with a Kolmogorov-Smirnov test in Fig. 3. The authors used a student t-test for assessment of differences in Fig. 6, but there is no description regarding t-test in the Methods section. It is not clear which statistics were used in Fig. 7.

Thanks, the text has been amended accordingly.

---

## [Decision Letter · Decision Letter 1]

27 Feb 2024

Epigenetic and physiological alterations in zebrafish subjected to hypergravity

PONE-D-23-33854R1

Dear Dr. Ribas,

We’re pleased to inform you that your manuscript has been judged scientifically suitable for publication and will be formally accepted for publication once it meets all outstanding technical requirements.

Kind regards,

Pierre Denise, Ph.D, M.D.

Academic Editor

PLOS ONE

Additional Editor Comments (optional):

Reviewers' comments:

Reviewer's Responses to Questions

**Comments to the Author**

1. If the authors have adequately addressed your comments raised in a previous round of review and you feel that this manuscript is now acceptable for publication, you may indicate that here to bypass the “Comments to the Author” section, enter your conflict of interest statement in the “Confidential to Editor” section, and submit your "Accept" recommendation.

Reviewer #1: All comments have been addressed

Reviewer #2: All comments have been addressed

2. Is the manuscript technically sound, and do the data support the conclusions?

Reviewer #1: Yes

Reviewer #2: Yes

3. Has the statistical analysis been performed appropriately and rigorously? 

Reviewer #1: Yes

Reviewer #2: Yes

4. Have the authors made all data underlying the findings in their manuscript fully available?

Reviewer #1: Yes

Reviewer #2: Yes

5. Is the manuscript presented in an intelligible fashion and written in standard English?

Reviewer #1: Yes

Reviewer #2: Yes

6. Review Comments to the Author

Reviewer #1: The authors have fully complied with this request and have revised the text and graphic and visual data.

Reviewer #2: The revised version bring additional information. The authors well addressed most of my concerns by revising the paper.

7. PLOS authors have the option to publish the peer review history of their article (what does this mean?). If published, this will include your full peer review and any attached files.

Reviewer #1: No

Reviewer #2: No

---

## [Editor Report · Acceptance letter]

26 Apr 2024

PONE-D-23-33854R1 

PLOS ONE

Dear Dr. Ribas, 

I'm pleased to inform you that your manuscript has been deemed suitable for publication in PLOS ONE. Congratulations! Your manuscript is now being handed over to our production team.

Kind regards, 

on behalf of

Pr. Pierre Denise 

Academic Editor

PLOS ONE